# Tetraspanin Cd9b and Cxcl12a/Cxcr4b have a synergistic effect on the control of collective cell migration

Katherine S. Marsay[1,2,3‡], Sarah Greaves[4‡], Harsha Mahabaleshwar[5], Charmaine Min Ho[5], Henry Roehl[4]*, Peter N. Monk[2], Tom J. Carney[3,5], Lynda J. Partridge[1]

1 Department of Molecular Biology and Biotechnology, University of Sheffield, Sheffield, United Kingdom, 2 Department of Infection, Immunity and Cardiovascular Science, University of Sheffield, Sheffield, United Kingdom, 3 Institute of Molecular and Cell Biology, Agency for Science, Technology and Research, Singapore, Singapore, 4 Department of Biomedical Science, University of Sheffield, Sheffield, United Kingdom, 5 Lee Kong Chian School of Medicine, Experimental Medicine Building, Yunnan Garden Campus, Nanyang Technological University, Singapore, Singapore

‡ KM and SG are joint first authors and contributed equally to this work.
* h.roehl@sheffield.ac.uk

**Data Availability Statement:** All relevant data are within the paper and its Supporting Information files.

## Abstract

Collective cell migration is essential for embryonic development and homeostatic processes. During zebrafish development, the posterior lateral line primordium (pLLP) navigates along the embryo flank by collective cell migration. The chemokine receptors, Cxcr4b and Cxcr7b, as well as their cognate ligand, Cxcl12a, are essential for this process. We corroborate that knockdown of the zebrafish cd9 tetraspanin orthologue, cd9b, results in mild pLL abnormalities. Through generation of CRISPR and TALEN mutants, we show that cd9a and cd9b function partially redundantly in pLLP migration, which is delayed in the cd9b single and cd9a; cd9b double mutants. This delay led to a transient reduction in neuromast numbers. Loss of both Cd9a and Cd9b sensitized embryos to reduced Cxcr4b and Cxcl12a levels. Together these results provide evidence that Cd9 modulates collective cell migration of the pLLP during zebrafish development. One interpretation of these observations is that Cd9 contributes to more effective chemokine signalling.

## Introduction

Cells can migrate individually or in groups, the latter is known as collective cell migration. During this process, cells exhibit coordinated behaviour, group polarisation and maintain cell-cell contacts [1]. This mode of migration is employed during embryonic development in the morphogenesis of multiple organ systems and is also important for effective immune responses. This mechanism contributes to several diseases including metastatic cancer and rheumatoid arthritis [2].

The zebrafish lateral line consists of a series of mechanosensory organs (neuromasts), which are distributed along the lateral surface of the zebrafish body and connected to the central nervous system by afferent axons [3]. This arrangement of innervated neuromasts is

**Funding:** The project was partly funded under the grant HR received from the MRC (MR/J001457/1). KM was funded by a studentship from IMCB-A-STAR Singapore and the University of Sheffield. The funders had no role in study design, data collection and analysis, decision to publish, or preparation of the manuscript.

**Competing interests:** The authors have declared that no competing interests exist.

achieved by the collective migration of a placodal primordium of approximately 100 cells, arising just posterior to the otic placode [4]. This posterior lateral line primordium (pLLP), then migrates along the horizontal myoseptum to the tip of the tail. During migration, clusters of cells are deposited from the trailing end of the pLLP, which ultimately differentiate into neuromasts.

Primordium migration is directed by expression of the chemokine Cxcl12a along the horizontal myoseptum, which is received by two chemokine receptors expressed within the primordium [5–7]. Cxcr4b is expressed in the leading two thirds of the primordium, where ligand binding induces Gβ1 signalling and actin polymerisation to promote a migratory phenotype [8]. Cxcr7 is expressed in the anterior-most third and functions as a ligand sink, allowing the formation of a local Cxcl12a gradient. This results in directed migration of the primordium along the Cxcl12a pathway towards the caudal fin [9, 10]. There is evidence that the membrane environment of chemokine receptors, including Cxcr4 and Cxcr7, strongly influences their signalling properties. This includes homodimerisation, oligomerisation and heteromerisation with other membrane receptors [11]. In addition, chemokine receptors have been linked to membrane microdomains where the lipid environment strongly modulates function [12].

Tetraspanins are a large family of small integral membrane proteins, which have been shown to organise neighbouring membrane proteins into complexes called tetraspanin enriched microdomains. This is often referred to as the tetraspanin web, as different interactions build to form dynamic signalling networks that often induce similar functional outcomes [13]. Thus, tetraspanins are associated with a wide variety of cellular functions including signalling and cell migration [14]. For example, the tetraspanin CD9 is downregulated in many human cancers including lung, breast and ovarian, and reduced CD9 expression is related to colon cancer metastasis [15]. In particular, CD9 has been shown to regulate, and be regulated by, CXCL12-CXCR4 signalling [16–18]. CD9 was found in close proximity to CXCR4 on the membrane of B acute lymphoblastic leukaemia cells in vitro, and enhanced their CXCL12 dependent migration [18]. If regulation of CXCL12/CXCR4 signalling by CD9 is conserved, we hypothesised Cd9 would be strongly expressed in the zebrafish pLLP, where it might modulate Cxcr4 signalling and thus pLLP migration. Indeed microarray experiments have localised the zebrafish *cd9b* paralogue to the migrating primordium, whilst a morpholino (MO) experiment indicated loss of *cd9b* altered primordium structure at 36 hours post fertilisation (hpf) and reduced neuromast number at 48 hpf [2].

In this study we aimed to investigate further the role of zebrafish CD9 orthologues in Cxcr4b-mediated pLLP migration, through use of genetic knockouts. We show expression of both *cd9a* and *cd9b* paralogues in the primordium and confirmed the abnormal pLL phenotype seen previously with *cd9b* knockdown. The *cd9b* mutants showed delayed pLLP migration but did not fully replicate *cd9b* morphant pLL phenotypes. We therefore generated *cd9a; cd9b* double knockouts (*cd9* dKOs). These showed both reduced migration of the pLLP and increased sensitivity to reduced levels of Cxcr4b and Cxcl12a, supporting a role for CD9 in regulation of Cxcr4 signalling.

## Methods

### Zebrafish maintenance

Adult wild-type zebrafish (WT), *Tg(-8.0cldnb:LY-EGFP)*[zf106] and *cd9a/b* mutants were housed and bred in a regulated 14:10 hour light: dark cycle under UK Home Office project licence 40/3459 in the Bateson Centre aquaria at the University of Sheffield or in the Singapore IMCB zebrafish facility under the Biological Resource Centre oversight with project license IACUC

140924. Zebrafish were raised under the standard conditions at 28˚C [19]. Ages are expressed as hours (hpf) or days (dpf) post fertilisation.

## Morpholino injection

Antisense morpholino oligonucleotides (MO) were obtained from GeneTools LLC and re-suspended in MilliQ $H_2O$ to give a stock concentration of 1 mM and injected in one-cell stage embryos. A Flaming/Brown micropipette puller was used to create micro-injection needles from borosilicate glass capillary tubes (0.5 mm inner diameter, Sutter). The PV800 Pneumatic PicoPump, as part of the micro-injection jig, was set up to release the required amount of injection material by adjusting the air pressure and air expulsion time. For the knockdown of *cd9b* two MOs were designed, a translation blocker with the following sequence (*cd9b* MO1): 5'-tttatgaggagaaacccaagactga-3' and a splice site blocker (*cd9b* i2e3) with the following sequence: 5'-aacccctgaacacagagaaacaaca-3', whilst the published mismatch MO was used 5'- tttccctgctgcttatacagcgatg -3' [20]. For knockdown of *cxcr4b* and *cxcl12a*, the following sequences were used respectively, 5'-aatgatgctatc gtaaaattccat-3' and 5'-ttgagatccatgtttgcagtgtgaa-3' [21].

## Zebrafish mutant production

*cd9b* mutants were created from WT embryos using transcription activator-like effector nucleases (TALEN) and maintained on an WT background. TALEN (ZGene Biotech Inc., Taiwan) were provided in a pZGB4L vector, targeting the *cd9b* sequence 5'-ttgctctttatcttca-3'. Injected embryos were outcrossed and sequenced to identify mutations. A frameshifting deletion mutation (c.42_49del) was selected that caused premature termination within the first transmembrane domain (*cd9b*[pg15] allele). CRISPR-Cas9 was used to create *cd9a* mutants using the gRNA sequence 5'-gagtgtatatcctcattgcgg-3', which targeted the 3rd exon encoding part of the second transmembrane domain. *cd9* dKO mutants were created by injecting the above *cd9a* gRNA and Cas9 RNA into *cd9b*[pg15] embryos. These fish were screened for germline transmission by sequencing and backcrossed to *cd9b*[pg15] mutants. An indel mutation deleting 4bp and inserting 8bp (c.180_187delinsTCGCTATTGTAT; *cd9a*[la61]) generated a frameshift mutation resulting in a premature stop codon in exon 3, which was predicted to truncate the protein before the large extracellular domain. Heterozygous fish of the same genotype were incrossed and adult F2 fish were genotyped to identify homozygous *cd9b*[pg15]; *cd9a*[la61] (*cd9* dKO).

## In situ hybridisation

Embryos were raised at 28˚C in petri dishes containing E3 solution. The E3 was changed daily, and any dead embryos removed. At 30–32 hpf, embryos were anaesthetised using tricaine and dechorionated before returned to fresh E3 solution. At the relevant timepoint embryos were fixed overnight at 4˚C using 4% (w/v) paraformaldehyde (Sigma-Aldrich, UK) in phosphate buffered saline (PBS). Embryos were washed in PBS/0.05% (v/v) Tween 20 (PBST), then put through a methanol/PBS series using 30%, 60% and 100% (v/v) methanol before being stored in 100% methanol (Sigma-Aldrich) at -20˚C. In situ hybridisation (ISH) was carried out as described by [22], except for the embryo digestion with proteinase K, for which 30–32 hpf embryos were digested with 10 mg/ml proteinase K at 20˚C for 22 min. Primers used for PCR generation of the in situ probes are given in Table 1 below. The protocol was conducted with the embryos in 1.5 ml microfuge tubes for the first two days, after which they were held in 12-well plates for staining before transferring back to microfuge tubes for storage. Stained embryos were stored in the dark in 80% (v/v) glycerol.

**Table 1. Primers for making ISH probes.**

| Probe | Primer name | Sequence (5'-3) |
|---|---|---|
| *claudin b* | Claudinb F | aaacgaaaaagcatggcatc |
| *claudin b* | Claudinb R | gaggctgtttcaaacgtggt |
| *cd9a* | CD9a F | gtcatattcgcggtcgaagt |
| *cd9a* | CD9a R | ctgcgagaacaacaaagcaa |
| *cd9b* | CD9b F | gttcgccacaagtgcctgat |
| *cd9b* | CD9b R | tacatgttactttctctccaaacaat |

### Time lapse imaging

Time-lapse recording was performed using an inverted Zeiss LSM700 Confocal microscope. All larvae were anaesthetised using 0.02% tricaine and then embedded on their side in 1% low melting agarose (Lonza) on a glass bottom dish (MalTEK Corporation) and covered with E3 supplemented with 0.02% tricaine. Separate Z-stack images covering the depth of the horizontal myoseptum were taken at specified intervals over a specified period and assembled into a final movie at a specified frame rate.

### Statistics

Data distribution was first assessed for normality using a D'Agostino-Pearson omnibus K2 normality test on the experimental residuals, as well as creating a histogram of residuals. For normally distributed data, an ANOVA with Dunnet's or Holms-Sidak multiple comparisons tests were used. For non-normally distributed data non-parametric tests, the Mann-Whitney U test or Kruskal-Wallis with Dunn's multiple comparisons test, were used.

## Results

### Both *cd9* zebrafish paralogues are expressed in the lateral line

Zebrafish possess two Cd9 paralogues, Cd9a [NP_997784] and Cd9b [NP_998593] which show 60% and 59% identity to human CD9 respectively and 63% identity to each other using multiple sequence comparison by log-expectation (MUSCLE) [23]. Whole mount in situ hybridisation (WISH) using probes against *cd9a* and *cd9b* demonstrated expression of both paralogues in the migrating primordium at 36 hpf (Fig 1A–1D). Expression of both *cd9a* and *cd9b* was also observed in recently deposited neuromasts and was retained in neuromasts until at least 5 dpf (Fig 1G–1L).

### *cd9b* morphant phenotype

To evaluate if there was a role for either paralogue in pLLP migration, we initially focused on *cd9b* as its role in the primordium had been previously reported [2]. Two different MOs were designed to target *cd9b*; *cd9b* MO1 was a translation blocking MO which targeted the 5' UTR of *cd9b* RNA and *cd9b* i2e3 was a splice blocking MO designed against the intron 2—exon 3 splice site. These MOs were injected independently into 1-cell stage embryos to ensure that they produced the same phenotype. Embryos injected with either *cd9b* MO1 or *cd9b* i2e3, but not the mismatch or uninjected embryos, showed a significant decrease in neuromasts deposited (Fig 2). For embryos injected with *cd9b* MO1, there was a decrease in the percentage of trunk length between the first and last neuromasts (Fig 2I). This suggests that the primordium is stalling prematurely or migrating more slowly. This recapitulates and expands on the work by Gallardo et al., 2010 which suggested a role for Cd9b in lateral line development [2].

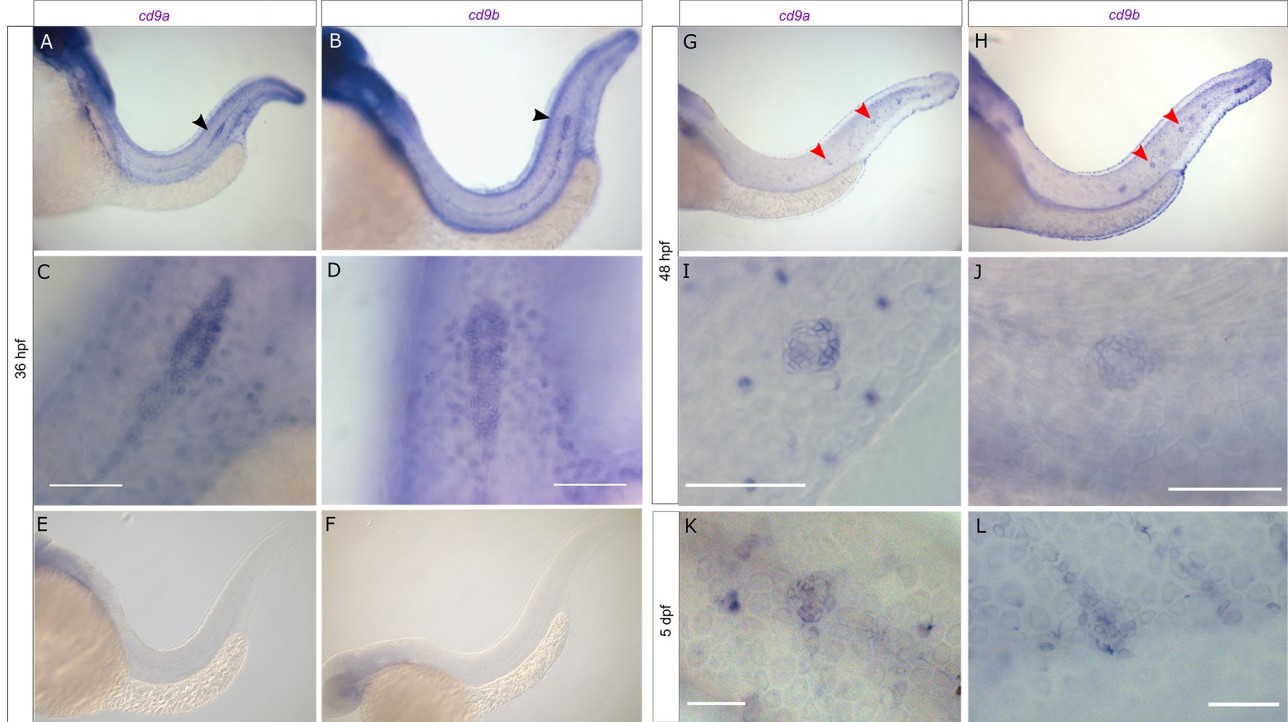

**Fig 1. *cd9a* and *cd9b* are expressed in the pLLP during zebrafish development.** A-F: Micrographs of WISH of *cd9a* and *cd9b* at 36 hpf with sense and anti-sense probes. (a-b) Overview shows staining in pLL; (c-d) higher resolution images show expression in the primordium; (e-f) sense probes show no staining. G-J: WISH of *cd9a* and *cd9b* at 48 hpf; (g-h) overview shows staining in pLL; (i-j) higher resolution images show expression in a neuromast. K-L: WISH of *cd9a* and *cd9b* at 5 dpf in a neuromast. Arrows indicate primordium (black) and neuromasts (red). Scale bar indicates 50 μm (white).

### *cd9b* mutant phenotype

To confirm this MO result, we created a TALEN mediated knockout of *cd9b*. A *cd9b* TALEN pair was designed by ZgeneBio using the program "TEL Effector Nucleotide Targeter 2.0". The TALEN pair was designed to target Exon 1 and predicted to cut in the 1st transmembrane domain (Fig 3A). The TALEN pair was injected and after 72 hpf a proportion of embryos were genotyped. Injected embryos showed clear mosaicism around the TALEN cut site so embryos were raised to create mosaic adult F0s. F0s transmitting a mutation to offspring were out-crossed with WT fish and the resulting F1 offspring were raised. To create a *cd9b* homozygous mutant line with a single mutant allele, F1s with a c.42_49del mutation were selected, incrossed and the resulting offspring raised to adulthood. This allele (*cd9b^{pg15}*) was used as it had the largest deletion and caused the earliest nonsense stop codon. The 8bp deletion in the *cd9b* allele leads to a frameshift in exon 1, changing codon 15 from TTT (Phe) to CAA (Glu), then 22 aberrant amino acids followed by a stop codon (p.Phe15GlufsTer22) (Fig 3B). *cd9b^{pg15}* homozygous mutants are viable and showed normal development. Loss of *cd9b* in situ signal in the *cd9b* mutants suggested strong nonsense mediated decay (NMD) of the mutated allele (S1 Fig). Due to the phenotypes seen in *cd9b* morphants, it was expected that a lateral line phenotype would be seen in *cd9b* mutants with fewer neuromasts deposited. However, no significant difference was found in the number of neuromasts at 52 hpf (Fig 3). As *cd9b* is expressed throughout pLL development, migration of the primordium as well as lateral line structure was assessed at 36 hpf. Although the pLL shows no structural abnormalities in *cd9b* mutants at 52 hpf, it is possible that *cd9b* mutants show a lateral line phenotype earlier in development

and have recovered by 52 hpf. At 36 hpf, *cd9b* mutants show delayed primordium migration, with the percentage of trunk traversed by the primordium reduced in mutants ([Fig 3]).

## Generation of a *cd9a* mutant

To ascertain a role for Cd9a, we targeted the *cd9a* gene using CRISPR/Cas9. A gRNA was designed to target *cd9a* in exon 3 ([Fig 4A]) and injected along with RNA encoding Cas9 into

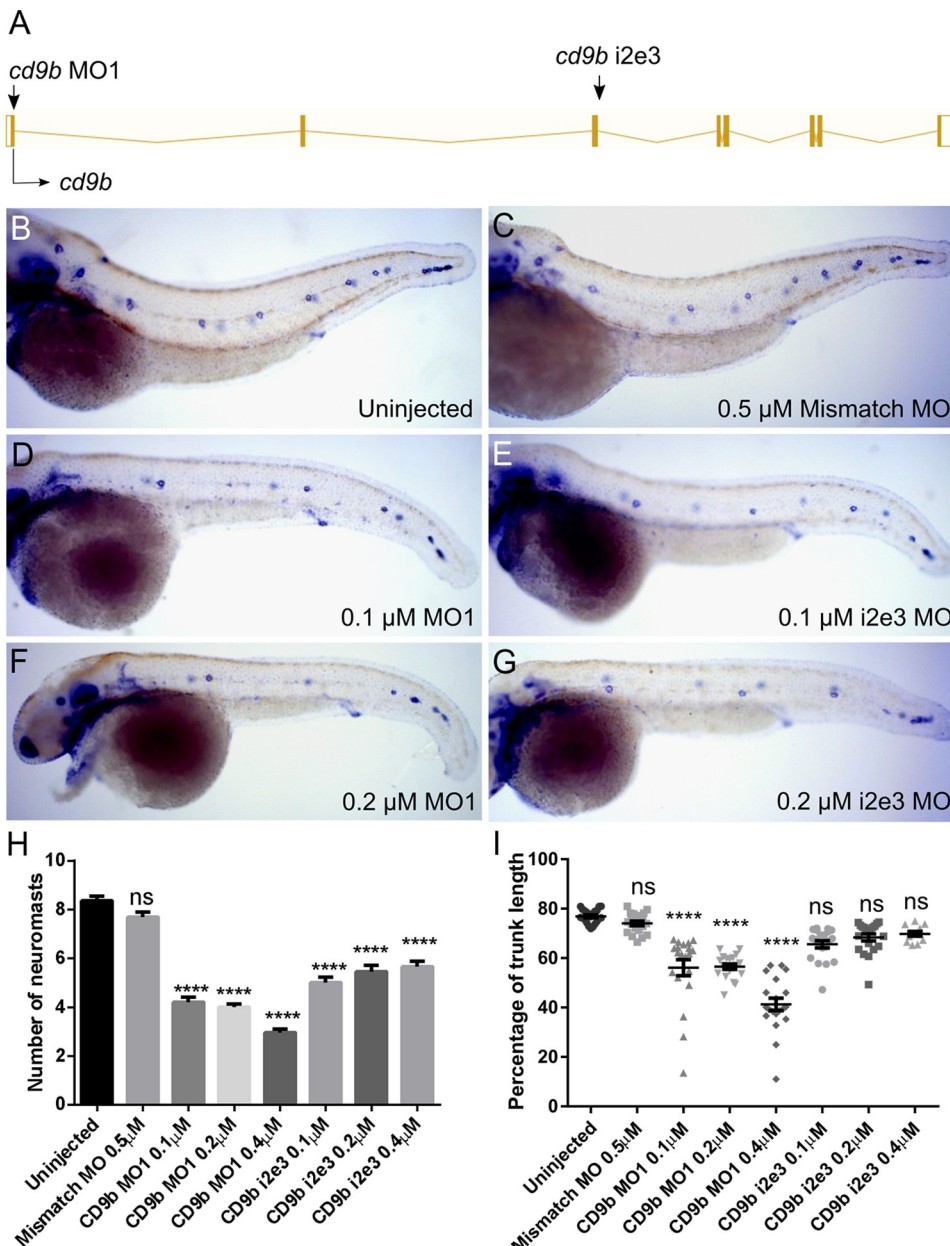

**Fig 2. *cd9b* MO caused a decrease in neuromasts deposited at 52 hpf.** A: Intron exon diagram of the *cd9b* gene with MO target sites. B-G: Representative images of WISH of *claudin b* at 52 hpf of (b) uninjected embryos, (c) embryos with 0.5 µM Mismatch, (d) 0.1 µM *cd9b* MO1, (e) 0.1 µM *cd9b* i2e3 MO1, (f) 0.2 µM *cd9b* MO1, (g) 0.2 µM *cd9b* i2e3 MO. H: Graph showing the number of neuromasts deposited at 52 hpf is significantly decreased after injection with both *cd9b* MO1 and *cd9b* i2e3 MO, with as low as 0.1 µM MO. I: Graph showing the percentage of trunk between the first and last deposited neuromast is decreased in *cd9b* MO1 morphants but not *cd9b* i2e3 morphants. ANOVA with Dunnett's multiple comparisons test was used. n = minimum 12 lateral lines analysed per condition. ****p = <0.05.

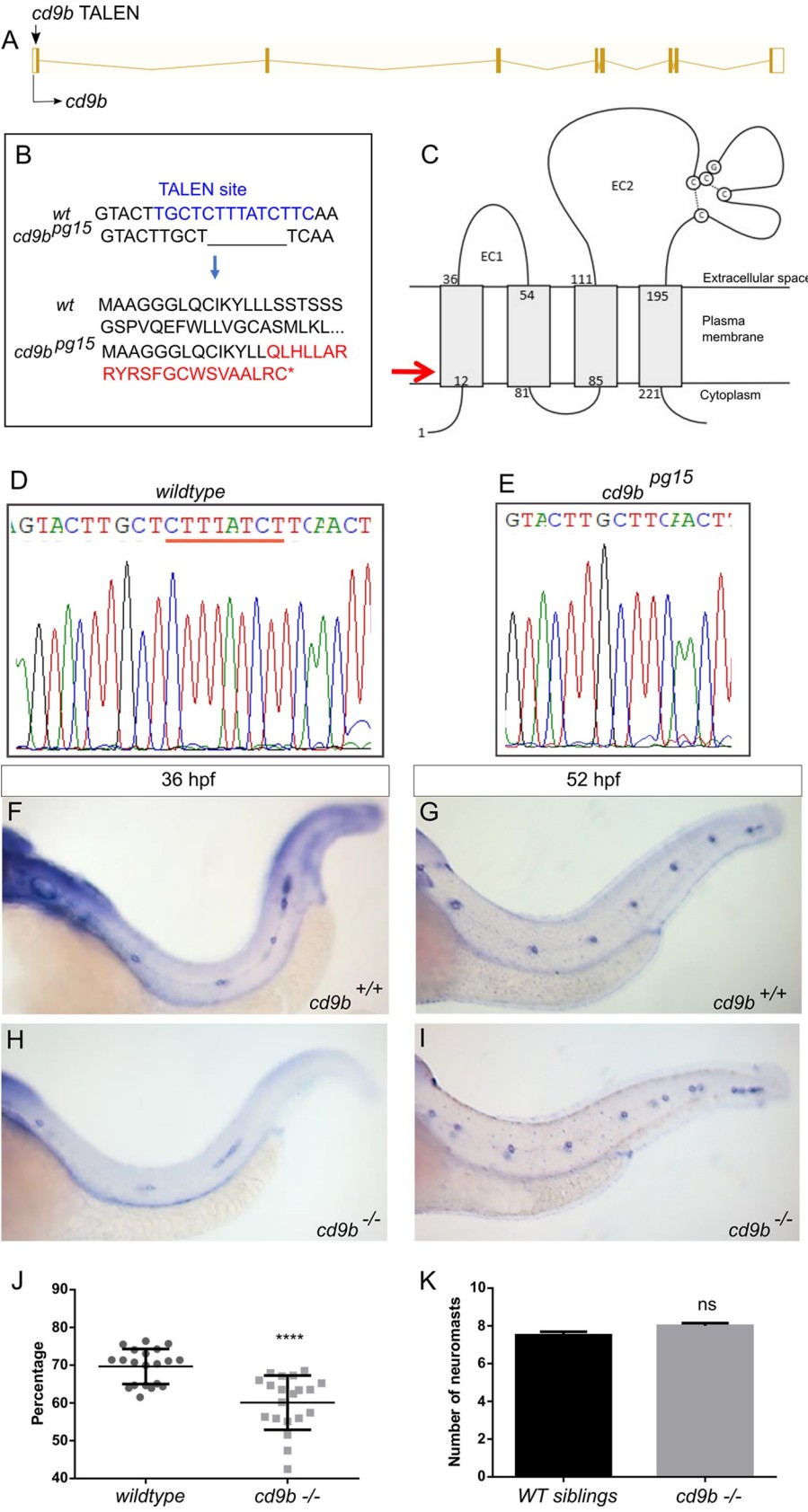

**Fig 3. *cd9b* mutant does not recapitulate morphant phenotype, although primordium migration is delayed at 36 hpf.** A: Nature of the *cd9b* mutant allele showing TALEN site location within the intron-exon structure of the gene. B: The TALEN target sequence in exon 1 is shown in blue; the 8bp deletion in the *cd9b* allele is indicated under the WT sequence as dashes. This leads to a frameshift changing codon 15 from TTT (Phe) to CAA (Glu), then 22 aberrant amino acids (red lettering) followed by a stop codon (*). C: Schematic of the Cd9b protein with location of premature stop codon given by red arrow. The disulfide bonds between the conserved CCG motif and conserved cysteines are indicated by the dashed lines. EC1/2 = Extracellular domain 1/2, aa = amino acid. D-E: Sequence chromatograms of genomic DNA from (d) WT and (e) *cd9b^pg15* alleles. Deleted base pairs are underlined in red. F-I: Representative images of WISH of *claudin b* ISH on (f-g) WT and (h-i) *cd9b* mutants at time shown. J-K: Graphs quantifying pLLP measurements in WT and *cd9b* KOs; (j) the migration of the *cd9b* KO primordium at 36 hpf is significantly delayed compared to WT. (k) There is no significance in number of neuromasts deposited at 52 hpf. Unpaired T test on untransformed data. n = minimum 20. **** = p = <0.05.

WT embryos to generate *cd9a* KOs. After screening for mosaicism, the CRISPR/Cas9 injected embryos were raised to maturity and screened for germline transmission of the *cd9a* mutation. F0s were outcrossed with WT fish to generate F1 *cd9a* single heterozygous mutants carrying an indel mutant allele (c.183_186delinsAT; *cd9a^ic62*). This indel led to a frameshift mutation, changing codon 62 from ATT (Iso) to TGC (Cys), then generating 54 aberrant amino acids followed by a premature stop codon before the EC2 domain (p.Iso62Cysfs54Ter) (Fig 4B and 4C). Heterozygous *cd9a^+/ic62* adults were in-crossed to produce the F2 generation. Adult F2s were viable and fertile and were genotyped to identify homozygous *cd9a^ic62* KOs. Loss of *cd9a* in situ signal in the *cd9a* mutants suggested strong NMD of the mutated allele (S2 Fig). The effect of *cd9a* KO on pLL development was investigated but overall, there were no major defects. No significant difference was found in the number of neuromasts deposited at 48 hpf or primordium migration at 36 hpf (Fig 4F–4K).

## Generation of a *cd9a; cd9b* double mutants

Due to the redundant nature of tetraspanins, and *cd9a* having a similar mRNA expression pattern to *cd9b*, it was speculated that functional redundancy between Cd9a and Cd9b might be masking stronger phenotypes in the single mutants. Hence, we generated *cd9a; cd9b* double mutants. To do this, the same *cd9a* gRNA as above (Fig 4) and *cas9* mRNA were injected into *cd9b^pg15* embryos. These fish were screened for mutation of *cd9a* as above and then backcrossed to *cd9b^pg15* mutants. An indel mutation in exon 3 was detected (c.180_184delinsTCGC TATTGTAT) that was predicted to lead to a frameshift mutation and an early stop codon, which truncated the protein before the EC2 domain (p.Leu61Alafs9Ter) (Fig 5A–5D). Transheterozygous fish were crossed and adult F2s were genotyped to identify doubly homozygous *cd9b^pg15; cd9a^la61* individuals, from now on referred to as *cd9* dKOs. Adult *cd9* dKOs were viable and fertile and loss of *cd9a* and *cd9b* in situ signal suggested strong NMD of the mutated alleles (S3 Fig). It was expected that with both paralogues knocked out, the lateral line phenotype would now be as severe as *cd9b* morphants with fewer neuromasts deposited. However, no significant difference was found in the number of neuromasts at 48 hpf (Fig 5). Primordium migration was significantly delayed in *cd9* dKO embryos at 36 hpf (Fig 5). This matches the phenotype shown by the *cd9b^pg15* single KOs. To see if the knockout of *cd9a^la61* had an additional effect on the primordium delay, embryos from *cd9b^pg15* KOs were compared to those from *cd9* dKOs (S4 Fig). The distance the primordium had migrated at 36 hpf was measured and no difference was seen between *cd9b^pg15* single KO and *cd9b* dKO.

In order to further investigate the migration and organisation of the primordium in *cd9* dKO embryos they were crossed into the *cldnb:gfp* transgenic line. Under the *claudin b* promoter GFP is expressed in the lateral line primordium and newly deposited neuromasts [5]. This allowed observation of the primordium migration in real time. Primordium migration

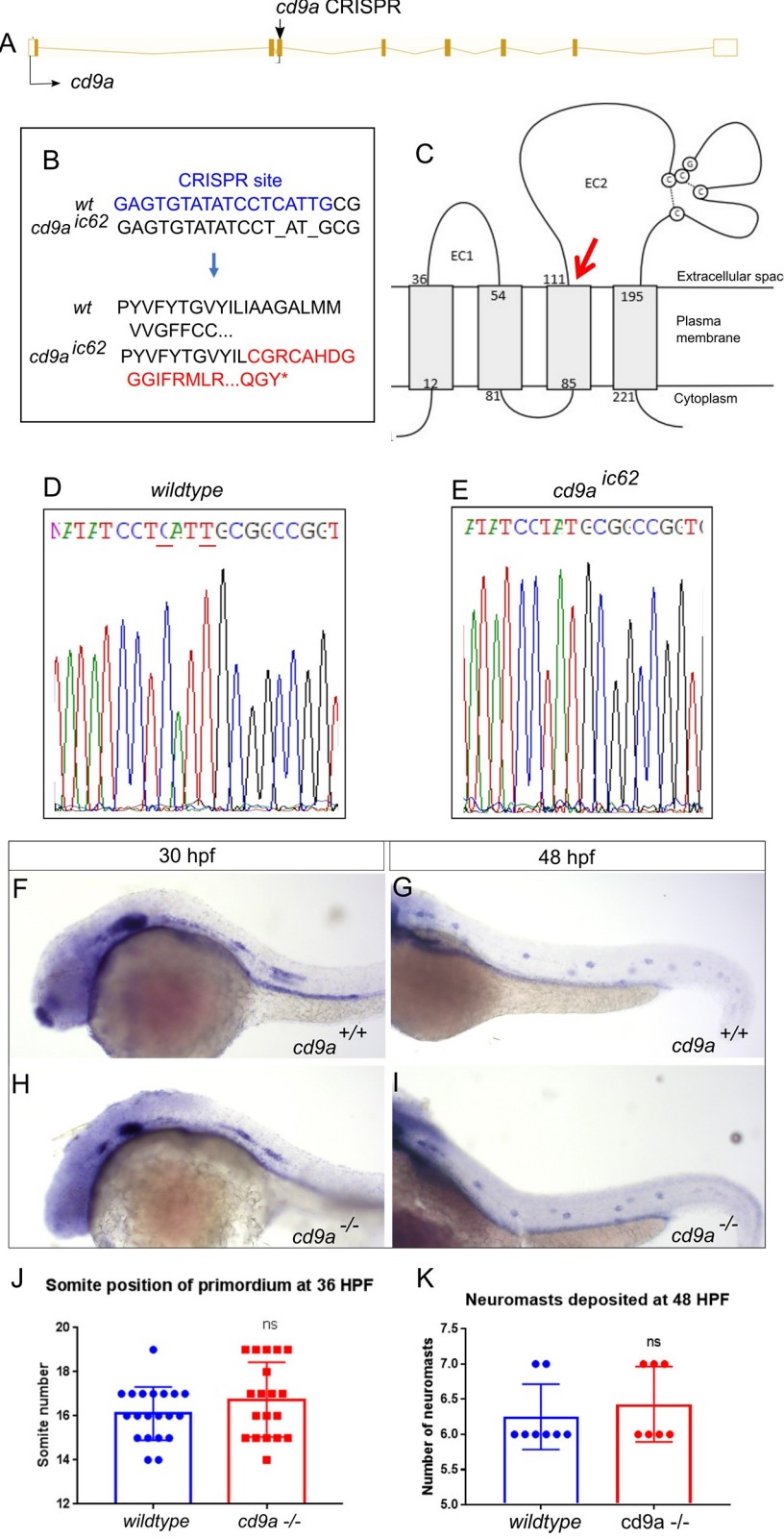

**Fig 4. Generation of *cd9a* mutant does not show any abnormal pLLP development.** A: Nature of the *cd9a* mutant allele showing CRISPR site location within the intron-exon structure of the gene. B: The CRISPR target sequence in exon 3 is shown in blue; the 2bp deletion in the *cd9a* allele is indicated under the WT sequence as dashes. This leads to

a frameshift changing codon 62 from ATT (Iso) to TGC (Cys), then 54 aberrant amino acids (red lettering) followed by a stop codon. C: Schematic of the Cd9a protein with location of premature stop codon, indicated by red arrow, at a predicted 115 aa. EC1/2 = Extracellular domain 1/2 D-E: Sequence chromatograms of genomic DNA from (d) WT and (e) *cd9a^{ic62}* allele. Deleted base pairs are underlined in red. F-G: *claudin b* expression in WT at (f) 30 hpf and (g) 48 hpf. H-I: *claudin b* expression in *cd9a* mutants at (h) 30 hpf and (i) 48 hpf. J-K: Graphs quantifying pLLP measurements in WT and *cd9a* KOs; (j) migration of the *cd9a* KO primordium at 36 hpf is similar to WT. (k) There is no significance in number of neuromasts deposited. Significance was assessed using an unpaired t test. N = minimum 7. Bars show mean +/- SD.

appeared normal in the *cd9* dKO embryos (Fig 6, S1 and S2 Videos). At the leading edge of the primordium filopodia could be seen as well as proliferating cells. Within the migrating primordium rearrangements occurred and rosettes were formed, increasing the length of the primordium. As the trailing cells decelerated and deposited cells as neuromasts, the primordium reduced in size. Quantification of various aspects of the primordium during deposition revealed similar results in WT and mutants (S5 Fig).

## Cd9 interacts with the Cxcr4b/Cxcl12a pathway in the migrating zebrafish lateral line primordium

To investigate if *cd9* had an effect on *cxcr4b* expression in the primordium, *cxcr4b* WISH was performed but no perturbation of *cxcr4b* expression was seen in the *cd9b* KOs (S6 Fig). Quantification showed no significant difference between the expression of *cxcr4b* in WT and *cd9b* KOs. To investigate if Cd9 was affecting migration in the zebrafish primordium through the Cxcr4b/Cxcl12a pathway, an experiment using MOs was conducted. *Cxcr4b* and *cxcl12a* MOs induce premature stalling of the primordium at concentrations of 1.5 mM and 0.5 mM respectively [21]. 100 μM of *cxcr4b* MO was found to be the highest concentration injected into WT embryos that did not induce a phenotype. However, this concentration of MO resulted in a further delay of the primordium in the *cd9* dKOs (Fig 7A). For *cxcl12a* the highest concentration of MO that did not induce a phenotype in WT embryos was determined to be 12.5 μM but this had no further effect on the primordium migration in *cd9* dKOs (S7 Fig). If the *cxcl12a* MO concentration was raised to 25 μM then the primordium in both WT and *cd9* dKOs was delayed although the primordium in *cd9* dKOs was significantly more delayed (Fig 7B).

## Discussion

In this study we investigated the role of Cd9 in the migration of the pLLP in zebrafish development. We first verified previous data showing *cd9b* expression in the primordium at 30 hpf [2]. We also noticed *cd9b* was expressed throughout recently deposited neuromasts at 48 hpf and 5 dpf. Our MO knockdown recapitulated previous results from Gallardo et al., 2010. *cd9b* morphants showed fewer neuromasts deposited, which suggested a role for Cd9b in the development of the lateral line. A homozygous *cd9b* mutant was created using TALENs but surprisingly, the morphant phenotype was not recapitulated in *cd9b* homozygous mutants. This lack of phenotype could be due to several reasons. Firstly, the phenotype seen in *cd9b* morphants could be due to off target effects of the MOs and not due to *cd9b* knockdown. However, the fact that two different MOs and researchers induced the same phenotype suggests it was not the lack of specificity [2]. Secondly, the Cd9b truncated protein could have some residual function, although ISH results showed a downregulation of mRNA suggesting NMD was occurring. Also, the *cd9b* mutation aborts the normal sequence after aa15, before the EC2 domain, so any translated protein would be expected to have minimal function. Finally, the *cd9b* morphant phenotypes may be specific but tetraspanin redundancy could rescue the phenotype in

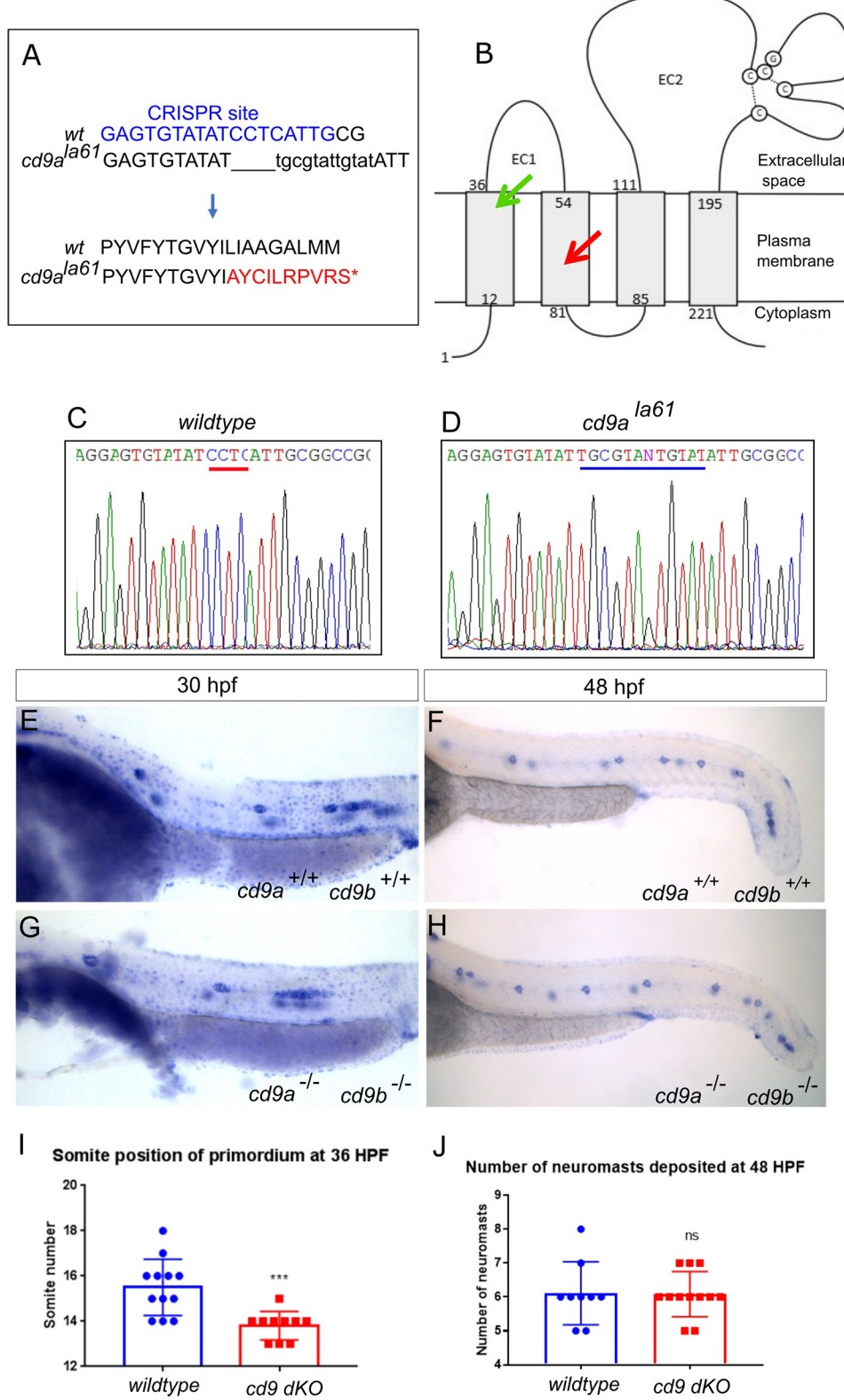

**Fig 5. Generation of *cd9a* mutant in the *cd9b* [pg15] KO background to create a double *cd9* KO mutant.** A: *cd9a* mutant allele sequence from exon 3, showing CRISPR target sequence in blue. The 4bp deletion (dashes) and 12bp insertion (lowercase) results in an 8bp insertion as indicated under the WT sequence. This leads to a frameshift

changing codon 61 from CTC (Leu) to GCG (Ala), then 9 aberrant amino acids (red lettering) followed by a stop codon. B: Schematic of the Cd9 protein with location of premature stop codon for Cd9a (red arrow) and Cd9b (green arrow) at a predicted 70 aa and 36 aa respectively. EC1/2 = Extracellular domain 1/2. C-D: Sequence chromatograms of genomic DNA from (c) WT and (d) *cd9a^la61* allele. Deleted base pairs are underlined in red and inserted base pairs in blue. E-F: *claudin b* expression in WT at (e) 30 hpf and (f) 48 hpf. G-H: *claudin b* expression in *cd9* dKO mutants at (g) 30 hpf and (h) 48 hpf. I-J: Graphs showing pLLP measurements in WT and *cd9* dKOs; (i) migration of the *cd9* dKO primordium at 36 hpf is significantly delayed compared to WT. (j) There is no significance in number of neuromasts deposited. Significance was assessed using an unpaired t test. N = minimum 9, p = <0.001, n1 = neuromast 1 etc. Bars show mean +/- SD.

*cd9b* mutants. Tetraspanins are well known for their redundancy within the tetraspanin family and mouse knockouts of single tetraspanins often appear healthy and viable with mild phenotypes, whereas double tetraspanin knockout mice often show increased numbers and severity of phenotypes [24–31]. In zebrafish, this redundancy may be amplified due to the occurrence of a fish-specific whole genome duplication in teleost fish. This means that many tetraspanins, for which there is one mammalian ortholog, have two paralogs in zebrafish [32–36]. This is true for CD9, as mammals have a single CD9 whereas zebrafish have Cd9a and Cd9b. We demonstrated *cd9a* to also be expressed in the pLL in a similar pattern to *cd9b*. Furthermore, NMD of mutated genes, as we have seen in both our mutants, has been shown to invoke an upregulation of closely related genes as a genetic compensation mechanism [37].

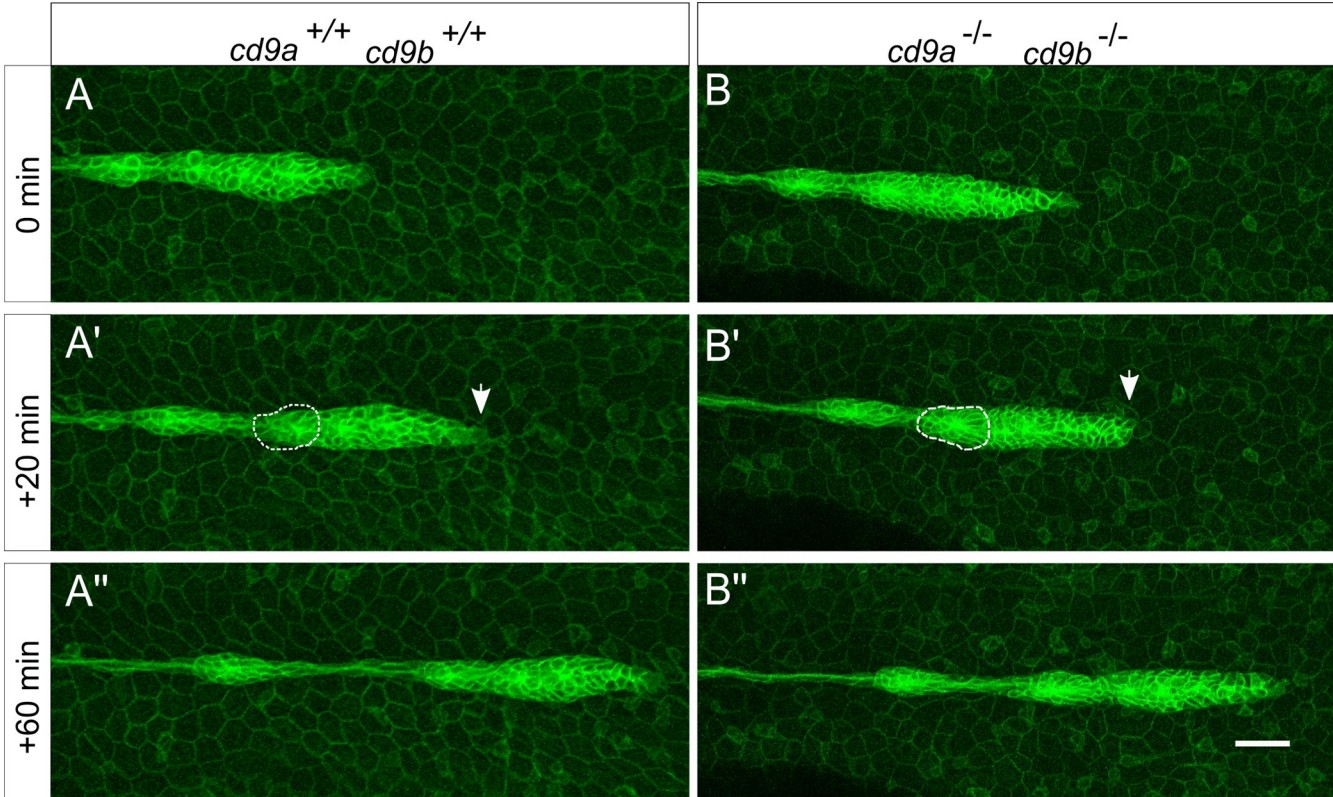

**Fig 6. Primordium organisation appears normal in *cd9* dKO(*cldnb:gfp*) embryos from 30 hpf.** A-B: Still images from a time-lapse recording of a migrating primordium in (a) WT and (b) *cd9* dKOs. 0 minutes shows initial deposition as a proneuromast becomes distinct from the primordium and then 2 sequential images show (a'-b') 20 minutes and (a"-b") 60 minutes later. In the primordium of both WT and *cd9* dKOs, filopodia can be seen at the leading edge (white arrow) and the formation of rosettes in the trailing edge (white dashed circle). Representative images from two videos that included two depositions each. Scale bar: 20 μm.

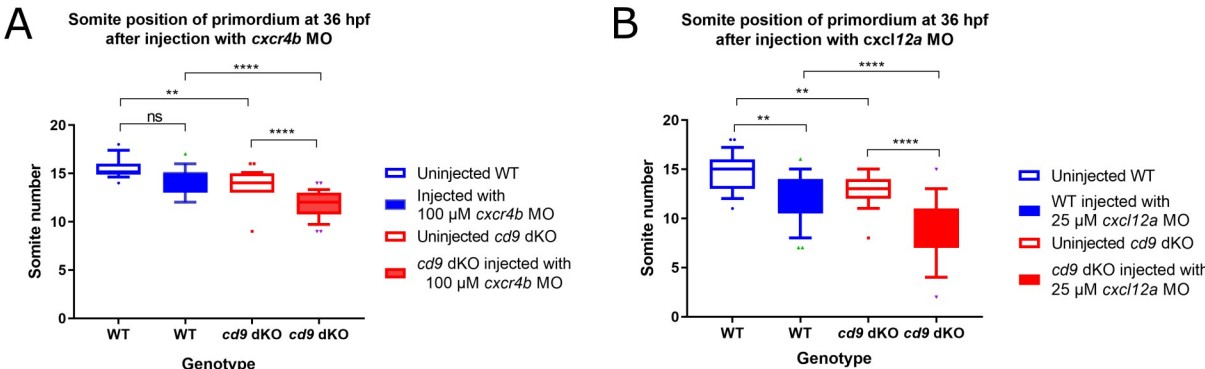

**Fig 7. *cxcr4b* and *cxcl12a* MOs further delay primordium migration at 36 hpf in *cd9* dKO embryos.** A-B: Distance migrated by the primordium (labelled by *claudin b* ISH) were recorded for (a) injection of 100 μM *cxcr4b* MO, and (b) 25 μM *cxcl12a* MO. Significance was assessed using one-way ANOVA, * = p<0.05, ** = p<0.01, *** = p<0.001 **** = p<0.0001. N = minimum 15. Box extends from the 25th to 75th percentile and whiskers from 10th to 90th.

Following this, tetraspanin *cd9a* was targeted for knockout using CRISPR technology. C*d9a*[ic62] homozygous mutants and *cd9b*[pg15] *cd9a*[la61] double homozygous mutants were generated. *Cd9a* KO embryos displayed a normal lateral line phenotype with normal primordium migration and neuromast deposition. Initial analysis of *cd9* dKO embryos using the same measurements revealed that the number of neuromasts deposited and spacing were normal yet the distance of primordium migration at 36 hpf was significantly delayed. It was expected that knockout of *cd9a* would result in a stronger pLL phenotype but *cd9* dKOs showed a similar phenotype to the *cd9b* KOs. Furthermore, when primordium migration of *cd9b* single KO was compared to that of the *cd9* dKOs there was no difference. These results suggest that the different phenotypes seen between the *cd9b* morphants and mutants are not due to compensation by *cd9a*. It remains possible that there is compensation by another tetraspanin [38]. Close relations to CD9 are the tetraspanins CD81 and CD63, which have been shown previously to substitute for CD9 in some circumstances. Overexpression of CD81 partially rescued the infertility phenotype seen in CD9 KO mice [39] and it was recently demonstrated that CD9 deletion in human melanoma cells was quickly compensated by CD63 expression upregulation [40]. It would be interesting to perform RNA-seq or SWATH-MS on the *cd9* dKOs to identify upregulated RNA and proteins respectively [41].

Videos generated of *cd9* dKOs with fluorescent *claudin b* showed primordium migration and internal organisation were normal. Measurements of the primordium after DAPI stain also revealed the primordium were similar sizes between mutants and WT at 36 hpf. Overall, it seemed the delay in migration at 36 hpf in the *cd9* dKO embryos was not due to alterations in the primordium organisation. Therefore, the *cd9* dKO phenotype appeared to be a migratory defect that could either be due to a change in the onset of migration, a change in the speed of migration or a combination of the two. CD9 has previously been reported to promote CXCR4b/CXCL12a signalling in mammalian cells so we decided to investigate a potential interference with this chemokine signalling pathway [16–18]. Results from the WISH analysis and quantification showed no obvious perturbation of *cxcr4b* expression in the *cd9* KOs. To postulate if there was some interaction between Cd9 and the Cxcr4b/Cxcl12a signalling pathway a MO experiment was performed. It was theorised that a primordium lacking Cd9 would be more susceptible to disruption by sub-functional doses of either *cxcr4b* or *cxcl12a* MO than WT. Injection with a sub-functional concentration of 100 μM *cxcr4b* MO demonstrated that the *cd9* dKO larvae were more sensitive to MO treatment compared to WT with significant

retardation in primordium migration in the dKO. This suggests that there is some interaction between Cd9 and Cxcr4b within the primordium that promotes migration. Injection with *cxcl12a* MO was more ambiguous as at a sub-functional concentration of 12.5 μM neither the WT nor the *cd9* dKO primordium migration was perturbed. When the dose of *cxcl12a* MO was increased to 25 μM a migratory delay was induced in both genotypes but the delay seen in the *cd9* dKO primordium was significantly worse. Despite disruption in the presence of the MO, the primordium was still able to eventually migrate the full length in the *cd9* dKO embryos, suggesting that Cd9 is not essential for maintaining Cxcr4b/Cxcl12a signalling but may play a regulatory or buffering role. A further experiment to check for interaction between Cd9 and Cxcl12a/Cxcr4b in the primordium would be to use immunohistochemistry to investigate co-localisation. Unfortunately, there are no tetraspanin, Cxcl12a or Cxcr4b antibodies available in zebrafish. However, two plasmids are available that encode *cxcr4b-egfp* or *cxcl12a-venus* constructs [42]. An interesting experiment would be to inject embryos with these plasmids and visualise the distribution of fluorescent versions of the proteins in the *cd9* dKOs.

We mentioned that the migratory defect could be due to a change in the onset of migration. This could be further investigated by performing *claudin b* ISHs at the onset and earliest stages of development of the posterior lateral line. On the other hand, to try and determine if Cd9b has a role in the speed of migration, ISHs could also be performed at various time points throughout primordium migration and speed could be calculated in a similar method to previous studies [43, 44].

We focused on a connection between Cd9 and Cxcr4b/Cxcl12 signalling in this paper because of previous research linking these proteins. However, Cd9, like other tetraspanins, contributes to cellular processes by organising molecules within the plasma membrane and there are many other signalling pathways involved in primordium development [45, 46]. The fundamental four are FGF, Wnt, Notch and chemokine signalling pathways and their specific and coordinated expression ensures the regular morphogenesis and migration of the pLL [47]. Even though the organisation of the primordium at high resolution appears normal, analysis of the expression of these crucial signalling molecules, through WISH or immunohistochemistry, during primordium development could reveal subtle changes in their organisation that may be affecting migration. Whilst Fgf signalling functions to determine rosette formation and deposition, Wnt signalling drives proliferation in the leading zone [46]. Proliferation analysis with BrdU incorporation could show if there are any changes in cell proliferation within the primordium, specifically in the leading edge. CD9 has also been demonstrated to downregulate expression of Wnt signalling pathways in cell culture [48, 49]. Another important component of collective cell migration is cell adhesion so migratory cells can pull insensitive cells in the same direction. Within the primordium cadherins mediate cell-cell adhesion between primordium cells and are required for directed and robust migration [50]. Cadherin-dependent intercellular adhesion regulated by tetraspanins plays an important role in tumour invasion and metastasis [51, 52]. CD9 has been associated with keratinocyte motility by regulating E-cadherin-mediated cell-cell contacts [53].

The use of computational models has been a valued tool to integrate analysis of the lateral line primordium acquired through microscopic, cellular, molecular and genetic analysis [47, 54]. These models allow researchers to edit parameters and observe their influences on the migrating primordium. This could help us choose the most appropriate avenue to explore next by checking what modifications are able to regenerate the phenotype we observe in the *cd9* dKO embryos.

In conclusion, we propose that within the primordium Cd9b is functioning to compartmentalise signalling molecules, like Cxcr4b, to orchestrate and amplify signalling upon ligand

binding but is not essential. This would explain why only subtle differences are seen between WT and *cd9* dKOs.

## Supporting information

**S1 Fig. *cd9b* is significantly decreased in *cd9b* KO embryos.** A-B: Representative images of *cd9b* WISH at 36 hpf in (a) WT and (b) *cd9b* homozygous embryos.
(TIF)

**S2 Fig. *cd9a* is significantly decreased in *cd9a* KO embryos.** A-D: Representative images of *cd9a* WISH on (a-b) WT and (c-d) *cd9a* homozygous embryos at time shown.
(TIF)

**S3 Fig. *cd9a* and *cd9b* are both significantly decreased in *cd9* dKO embryos.** A-B: Representative images of *cd9a* ISH on (a) WT and (b) *cd9* dKO mutants at time shown. C-D: Representative images of *cd9b* ISH on (c) WT and (d) *cd9* dKO mutants at time shown.
(TIF)

**S4 Fig. *cd9a* is not compensating for *cd9b* in the primordium.** Graph showing the distance migrated by the primordium at 36 hpf is the same in *cd9b* KO and *cd9* dKO embryos. Significance was assessed using an unpaired t test, N = minimum 17. Bars show mean +/- SD.
(TIF)

**S5 Fig. Primordium shape is normal in the *cd9* dKO embryos at 36 hpf.** A: Representative image of a DAPI-stained primordium from a WT zebrafish at 36 hpf with red outline to show measured area. B-E: Graphs showing measurements of the primordium (b) height, (c) width, (d) width to height ratio, and (e) area in WT and *cd9* KO embryos. The height was measured three times at equal points along the primordium and then averaged. The width of primordium was measured from the two furthest points along the middle of the primordium. Width was then divided by height to generate a ratio. The area was circled using the freehand selection and measured. Cells were counted using the multipoint tool on Image J software. Significance was assessed using an unpaired t test, p = <0.05, N = minimum 11. Bars show mean +/- SD.
(TIF)

**S6 Fig. *cxcr4b* expression is not altered in the pLLP of *cd9* dKO embryos at indicated stages.** A-D: Representative images of *cxcr4b* ISH in the primordium of (a,c) WT and (b,d) *cd9* dKO mutants at time shown. Scale bar: 50 μm. E-F: Graphs showing measurements of (e) length and (f) height of *cxcr4b* expression in the primordium of WT and *cd9* KO embryos at indicated stages. Length measurements were taken along the middle of the embryo between the two furthest points of expression within the primordium. Height measurements were taken between the two highest points of expression within the primordium. Significance was assessed using an unpaired T test. N = minimum 13 for 24 hpf and 11 for 30 hpf. Bars show mean +/- SD.
(TIF)

**S7 Fig. 12.5 μM of *cxcl12a* MO does not further delay primordium migration at 36 hpf in the *cd9* dKO embryos.** Distance migrated by the primordium (labelled by *claudin b* ISH) was recorded for injection of 12.5 μM *cxcl12a* MO. Significance was assessed using one-way ANOVA. N = minimum 13. Box extends from the 25th to 75th percentile and whiskers from 10th to 90th.
(TIF)

**S1 Video. WT primordium migration.** *WT*(*cldnb*:*gfp)* from 30–38 hpf. Images taken every 1 minute with a 20x objective. Movie length: 420 min.
(MP4)

**S2 Video. *cd9* dKO primordium migration.** *cd9* dKO(*cldnb*:*gfp)* from 30–36 hpf. Images taken every 1 min with a 20x objective. Movie length: 360 min.
(MP4)

## Acknowledgments

The Bateson Centre aquarium at the University of Sheffield, the zebrafish facility at IMCB Singapore, Dr Freek van Eden for his advice on creating zebrafish knockouts and Kunal Chopra for his support raising the TALEN line.

## Author Contributions

**Conceptualization:** Tom J. Carney, Lynda J. Partridge.

**Data curation:** Katherine S. Marsay, Sarah Greaves.

**Formal analysis:** Katherine S. Marsay, Sarah Greaves.

**Funding acquisition:** Henry Roehl.

**Investigation:** Harsha Mahabaleshwar, Charmaine Min Ho.

**Project administration:** Tom J. Carney, Lynda J. Partridge.

**Resources:** Henry Roehl, Tom J. Carney, Lynda J. Partridge.

**Supervision:** Henry Roehl, Peter N. Monk, Tom J. Carney, Lynda J. Partridge.

**Visualization:** Katherine S. Marsay, Sarah Greaves.

**Writing – original draft:** Katherine S. Marsay.

**Writing – review & editing:** Katherine S. Marsay, Tom J. Carney, Lynda J. Partridge.

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
