## [Decision Letter · Decision Letter 0]

9 Jul 2021

PONE-D-21-17584

CD9 tetraspanins convey robustness to CXCR4b signalling during collective cell migration

PLOS ONE

Dear Dr. Roehl,

Thank you for submitting your manuscript to PLOS ONE. After careful consideration, we feel that it has merit but does not fully meet PLOS ONE’s publication criteria as it currently stands. Therefore, we invite you to submit a revised version of the manuscript that addresses the points raised during the review process.

Your manuscript has been seen by two expert reviewers in the field. One reviewer highlights a shortcoming of the figures uploaded to the PLoS One server. Please check that the uploaded figures represent the final versions and correspond to the respective manuscript text. Please also carefully take into consideration the comments made by Reviewer 2, in terms of interpretation of your findings and the contribution of FGF signaling.

We look forward to receiving your revised manuscript.

Kind regards,

Christoph Winkler, Dr.

Academic Editor

PLOS ONE

Journal Requirements:

Reviewers' comments:

Reviewer's Responses to Questions

**Comments to the Author**

1. Is the manuscript technically sound, and do the data support the conclusions?

Reviewer #1: No

Reviewer #2: Partly

2. Has the statistical analysis been performed appropriately and rigorously? 

Reviewer #1: N/A

Reviewer #2: Yes

3. Have the authors made all data underlying the findings in their manuscript fully available?

Reviewer #1: No

Reviewer #2: Yes

4. Is the manuscript presented in an intelligible fashion and written in standard English?

Reviewer #1: Yes

Reviewer #2: Yes

5. Review Comments to the Author

Reviewer #1: The manuscript contains plenty of errors and most of the figures do not show all images/panels described in the text. At this stage, the manuscript is not competent for publication and should be prepared again with more attention.

Reviewer #2: The paper by Marsay et al revisits the role of the tetraspanin CD9 in the primordium. A previous microarray analysis study in 2010 had shown that CD9 is expressed in the primordium and that that CD9 morpholino injected embryos have delayed migration and fewer neuromasts. This study examines expression of cd9b and the closely related cd9a with whole mount in situ hybridization, confirming expression of these cd9 paralogues in the primordium. It recapitulates previous results showing that cd9 knockdown delays migration and reduces the number of neuromasts. It goes on to show that a TALEN generated mutant, predicted to have loss of CD9b function, does not recapitulate the morphant phenotype and though migration is delayed the number of neuromasts is not altered. The authors go on to determine if the difference between the morphant and mutant phenotype is related to the redundant role of cd9a by generating a CRISPR based mutant expected to have of function. However, the phenotype of the double mutant is not significantly different from that of cd9b alone. Finally, the authors explore the potential role of CD9 in chemokine dependent migration in the primordium by showing that a subcritical knock down of cxcr4b or cxcl12a results in slowing of migration only in the background of a double mutant loss of cd9b and cd9a.

Overall, though the study documents their results effectively, the observations only add modestly to what was previously known about cd9b function. Beyond showing that, unlike morphants, mutants do not have a reduction in the number of neuromasts, they also show that the difference between cd9b morphant and mutant phenotypes cannot be accounted for by redundant function of cd9a.

The authors demonstrate a reduction in primordium migration specifically in double cd9b/cd9a mutants but not wild-type embryos following partial knock down of cxcr4b or cxcl12a. They interpret this to suggest cd9 functions to make cxcr4b function more robust. While strictly speaking it may be correct to say the function is less “robust”, the analysis is minimal and it is remains unclear what the role of cd9 in chemokine might be or whether the exaggeration of deficits seen specifically in the double mutants necessarily result from a role of cd9 in determining efficacy of chemokine signaling. The deficits seen in cd9 double mutants are not typical of those seen when chemokine signaling alone is compromised as it is not even clear whether primordium migration is slower or simply delayed in its initiation. A deficit in chemokine signaling alone might be expected to slow migration and reduce spacing between deposited neuromasts, however, this is not what is seen in the mutants. Furthermore, as collective migration of the primordium is not just dependent on chemokine signaling but also, at a minimum, on Fgf signaling, it is not clear if the observed deficits in migration seen specifically in cd9 mutants might also be seen when Fgf signaling dependent migration is compromised. The authors should test if compromised migration, observed in cd9 double mutants in the background of compromised cxcr4b/cxcl12a function, is also seen when migration is slowed with subcritical doses of an Fgf signaling inhibitor. Are synergistic effects on migration only seen following manipulation of chemokine dependent migration and not following subcritical interference with Fgf-dependent migration? The literature suggests quite a broad range of potential roles for Cd9 and it would be premature to conclude that changes following loss of cd9b function demonstrated in this study are necessarily related to a role for CD9 in facilitating chemokine signaling.

6. PLOS authors have the option to publish the peer review history of their article (what does this mean?). If published, this will include your full peer review and any attached files.

Reviewer #1: No

Reviewer #2: No

---

## [Author Response · Author response to Decision Letter 0]

2 Oct 2021

Christoph Winkler, Dr.

Academic Editor

PLOS ONE

Manuscript no. PONE-D-21-17584

Manuscript title: Augmentation of chemokine signalling by CD9 tetraspanins facilitates collective cell migration

Dear Dr Winkler,

We were glad to learn that the reviewers found our work interesting and that, pending revisions, the manuscript is acceptable for publication in PLOS ONE. We modified our manuscript according to the comments of the reviewers as detailed in the point-by-point response below. The modified manuscript has highlighted modified text in red. We thank the reviewers for their constructive criticisms and hope that you will find the manuscript suitable for publication in PLOS ONE.

Sincerely,

Dr. Katherine Marsay

Detailed response to comments:

We would like to thank the reviewers for their constructive comments that helped us to improve the manuscript. As detailed below we modified the manuscript according to these comments. 

Journal Requirements

Comment: 1. Please ensure that your manuscript meets PLOS ONE's style requirements. 

Answer: High attention was made to ensure all PLOS ONE style requirements were met including: correct formatting in corresponding author and joint author information, correct acknowledgements, correct reference format.

Comment: 2. PLOS requires an ORCID iD for the corresponding author in Editorial Manager on papers submitted after December 6th, 2016. 

Answer: HR has validated his ORCID iD. 

Comment: 3. We note that you have included the phrase “data not shown” in your manuscript. Unfortunately, this does not meet our data sharing requirements. 

Answer: Previous results referred to as “data not shown” have been added to Supplementary figures S5 and S6. 

Reviewer #1:

Comment: The manuscript contains plenty of errors and most of the figures do not show all images/panels described in the text. 

Answer: All figures were evaluated in detail and corrected to match figure legends. Gene and protein labels were corrected.

Comment: Throughout the manuscript, the use of upper and lower case letters to label protein symbols for fish and for human should be corrected according to standard gene/protein nomenclature. 

Answer: All fish and human gene and protein labels have been corrected according to standard gene/protein nomenclature. 

Comment: reference format e.g. line 34

Answer: All references were checked and reformatted correctly. 

Comment: There were a few comments regarding figure 1 so I will address them together:

Image in A needs a different scale bar as it appears that the image in D is of different magnification than image in A. One image of the sense probes of cd9a and cd9b should be added. The embryo images in B and E appear younger than 48 hpf. It would be more demonstrative if authors can show alternative images for this time point. Although the image in A shows the labelling of primordium structure, a colocalization of cd9 in situ with a known marker of neuromast (such as ath1 or cldnb) would strongly support the author’s statement that cd9 genes are expressed in deposited neuromasts.

Answer: Figure 1 was redrawn to best illustrate cd9a and cd9b expression in the lateral line. For 36 hpf and 48 hpf a low and high magnification image is included so the expression can be seen in the posterior lateral line as a whole, as well as in detail in the primordium or neuromasts. The images of the neuromasts at high magnification show more clearly how cd9 is specifically expressed in the cell membranes of most cells in a deposited neuromast and supports the statement that both cd9 paralogues are expressed in deposited neuromasts. Scale bars are consistently 50 µM throughout the figure. As requested, one image of the sense probes of cd9a and cd9b were added.

Reviewer #2:

Comment: While strictly speaking it may be correct to say the function is less “robust”, the analysis is minimal and it is remains unclear what the role of cd9 in chemokine might be or whether the exaggeration of deficits seen specifically in the double mutants necessarily result from a role of cd9 in determining efficacy of chemokine signaling. 

Answer: Our morpholino results show that a lack of Cd9 exacerbates cxcr4b knockdown and primordium delay. The reviewer is correct that we cannot assign a specific role of Cd9 to Cxcr4b in this pathway. We changed some wording throughout our manuscript to support our results while keeping the potential mechanisms of Cd9 broad. e.g. 

New title: Augmentation of chemokine signalling by Cd9 tetraspanins facilitates collective cell migration. 

Abstract: Together these results provide evidence that Cd9 modulates collective cell migration of the pLLP during zebrafish development, potentially through promoting Cxcxr4b signalling.

Comment: The deficits seen in cd9 double mutants are not typical of those seen when chemokine signaling alone is compromised as it is not even clear whether primordium migration is slower or simply delayed in its initiation. A deficit in chemokine signaling alone might be expected to slow migration and reduce spacing between deposited neuromasts, however, this is not what is seen in the mutants.

Answer: We did not wish to make the claim that Cd9 is acting solely on Cxcr4 signalling and apologize if this was not made clear. We have now been more explicit about this in the discussion and explained that we chose to investigate Cxcr4b/Cxcl12 signaling as CD9 has been previously reported to interact and regulate with this signaling pathway in the migration of mammalian cells. 

Comment: Furthermore, as collective migration of the primordium is not just dependent on chemokine signaling but also, at a minimum, on Fgf signaling, it is not clear if the observed deficits in migration seen specifically in cd9 mutants might also be seen when Fgf signaling dependent migration is compromised. The authors should test if compromised migration, observed in cd9 double mutants in the background of compromised cxcr4b/cxcl12a function, is also seen when migration is slowed with subcritical doses of an Fgf signaling inhibitor. Are synergistic effects on migration only seen following manipulation of chemokine dependent migration and not following subcritical interference with Fgf-dependent migration? 

Answer: The reviewer is correct that we failed to discuss other potential mechanisms of Cd9 inhibition that could result in a delayed migration. We expanded the discussion to review other signalling mechanisms that could be relevant to Cd9 such as the regulation of Wnt signalling pathways. Cd9 has been previously reported to downregulate this pathway in cell culture. Unfortunately, I am no longer in a position to perform experiments on the mutant fish but the reviewer provoked an interesting discussion for future investigation.

Comment: The literature suggests quite a broad range of potential roles for Cd9 and it would be premature to conclude that changes following loss of cd9b function demonstrated in this study are necessarily related to a role for CD9 in facilitating chemokine signaling.

Answer: The reviewer is correct that we failed to discuss other mechanisms aside from chemokine signalling that could be associated with tetraspanins such as Cd9. We expanded the discussion to review other tetraspanin related functions that could be affecting cell migration in the developing primordium such as Cadherin-dependent intercellular adhesion.

---

## [Decision Letter · Decision Letter 1]

3 Nov 2021

PONE-D-21-17584R1Augmentation of chemokine signalling by CD9 tetraspanins facilitates collective cell migrationPLOS ONE

Dear Dr. Roehl,

Thank you for submitting your manuscript to PLOS ONE. After careful consideration, we feel that it has merit but does not fully meet PLOS ONE’s publication criteria as it currently stands. Therefore, we invite you to submit a revised version of the manuscript that addresses the points raised during the review process.

As you will see from the reviewers' comments below, both reviewers agree that the manuscript has improved with the revisions made. On the other hand, they both still feel that title and parts of the abstract overstate the findings made in this study and request to rephrase your wording. I agree with both that the experimental evidence provided is not sufficient to directly link Cd9 function to expression or activity of Cxcr4b. Other interpretations of your findings, as outlined by reviewer 2, are possible. Both reviewers suggest alternative titles and text changes. I recommend that you consider their proposed changes. In your response, please also address the comment raised by reviewer 2 on the movies of LLP migration, whether a delayed start of the LLP is possible in mutants. In your response letter, please also describe the contributions of the newly added co-authors.

We look forward to receiving your revised manuscript.

Kind regards,

Christoph Winkler, Dr.

Academic Editor

PLOS ONE

Journal Requirements:

Reviewers' comments:

Reviewer's Responses to Questions

**Comments to the Author**

1. If the authors have adequately addressed your comments raised in a previous round of review and you feel that this manuscript is now acceptable for publication, you may indicate that here to bypass the “Comments to the Author” section, enter your conflict of interest statement in the “Confidential to Editor” section, and submit your "Accept" recommendation.

Reviewer #1: All comments have been addressed

Reviewer #2: (No Response)

2. Is the manuscript technically sound, and do the data support the conclusions?

Reviewer #1: Partly

Reviewer #2: Yes

3. Has the statistical analysis been performed appropriately and rigorously? 

Reviewer #1: Yes

Reviewer #2: Yes

4. Have the authors made all data underlying the findings in their manuscript fully available?

Reviewer #1: Yes

Reviewer #2: Yes

5. Is the manuscript presented in an intelligible fashion and written in standard English?

Reviewer #1: Yes

Reviewer #2: Yes

6. Review Comments to the Author

Reviewer #1: This manuscript aimed to investigate the function of Cd9 tetraspanins in pLLP migration and neuromast formation in zebrafish. First, the authors showed that knocking down of cd9b by morpholinos delayed pLLP migration and reduced the number of deposited neuromasts along the trunk. This observation confirmed the data previously shown by Gallardo et al. To confirm the observed phenotypes the authors went on to use cd9b mutant. Unexpectedly, cd9b mutant only partly recapitulated the phenotypes seen in morphants in which the pLLP migration was transiently delayed, while the number of neuromasts was not affected in the mutant. The authors also looked into the phenotypes of cd9a mutant and cd9a/cd9b double mutants. While no obvious phenotype related to pLLP migration and neuromast formation was seen in cd9a mutant, the double mutant showed similar phenotype as of cd9b single mutant. The different phenotypes of cd9b morphants and cd9b mutant were explained that there could have compensations by other members of the tetraspanin family.

In another attempt, the authors injected morpholinos blocking Cxcr4b and Cxcl12a which previously known to interfere with the primodium migration, into cd9 double mutants. The authors showed that even the low concentrations of morpholinos, which had been known to cause no effect on pLLP migration, could induce further delay of pLLP migration in cd9 double mutants.

Overall, although the new findings in this study is not abundant, the manuscript provides certain clues for future research on the potential interplays among members of tetraspanin family, and between cd9b with Cxcl12/Cxcr4 pathway in the control of primodium migration and neuromast formation.

Minor comments:

The title “Augmentation of chemokine signalling by Cd9 tetraspanins facilitates collective cell migration” is missleading as there is no direct data showing that Cd9 enhances Cxcl12/Cxcr4 expresion or activity which in turn facilitates collective cell migration. The authors may consider a more direct title, for instance “Cd9b tetraspanin and Cxcl12a/Cxcr4b convey synergistic effect on the control of collective cell migration”.

Line 311: typo “Cxcrb” change to “Cxcr4b”

Reviewer #2: Marsay et al have resubmitted a revised version of their original manuscript “CD9 tetraspanins convey robustness to CXCR4b signaling during collective cell migration” with a new title “Augmentation of chemokine signaling by CD9 tetraspanins facilitates collective cell migration”.

A primary issue raised with the original submission was that, with limited analysis of deficits following morpholino, TALEN and/or CRISPR manipulation of cd9a and cd9b function in the zebrafish, the authors were not in position to definitively link Cd9 function to chemokine signaling. No new experiments help to establish a stronger functional relationship between cd9 and chemokine signaling in the primordium and the authors have addressed this weakness of the paper by toning down the suggested link, which is an improvement. However, both the title of the paper and some of the text still in my mind would suggest to the casual reader of the paper a stronger link than has been established.

What is not in doubt is that partial knock down of the cxcr4b in a cd9 double KO has a significant effect on retarding the position of the primordium compared to a situation, while the same subcritical dose of morpholino has no obvious effect in a wild-type background. This confirms that while the deficit in cxcr4b caused by the morpholino alone is not adequate to slow migration, an additional deficit caused by the loss of cd9 function results in delayed migration. Such observations on their own are not adequate to further conclude that the exaggerated deficits in migration resulting from the combined interference with cxcr4b or cxcl12a with cd9 necessarily result from a role of cd9 in the chemokine signaling pathway when it is known that multiple signaling systems in combination with chemokines contribute to primordium migration. The authors do raise this issue in the discussion, however the title “Augmentation of chemokine signaling by CD9 tetraspanins…” still suggests a stronger link between cd9 and chemokine signaling than the authors have evidence for in the paper and I think both the title and the abstract should be further modified to prevent unintentionally perpetuating a potentially erroneous conclusion based on limited analysis.

The title should be changed. A subtly different but perhaps less misleading title could be:

“Loss of Cd9 exaggerates deficits in collective migration caused by interference with chemokine signaling in the posterior Lateral Line primordium”

Similarly, the abstract should be edited to simply remove or edit the last line:

Loss of both Cd9a and Cd9b sensitized embryos to reduced Cxcr4b and Cxcl12a levels.

Together these results provide evidence that Cd9 modulates collective cell migration of the

pLLP during zebrafish development.”

OR

Loss of both Cd9a and Cd9b sensitized embryos to reduced Cxcr4b and Cxcl12a levels.

Together these results provide evidence that Cd9 modulates collective cell migration of the

pLLP during zebrafish development. One interpretation of these observations is that Cd9 contributes to more effective chemokine signaling.

One additional issue that requires comment. The video of pLLP migration does not show any obvious deficits in migration and migration speed was never assessed or reported. Is it possible that the difference in position of primordium observed by the authors results from a delay in the start of migration. Do the authors have evidence that argues against this? If not, this possibility should also be raised in the discussion.

7. PLOS authors have the option to publish the peer review history of their article (what does this mean?). If published, this will include your full peer review and any attached files.

Reviewer #1: No

Reviewer #2: No

---

## [Author Response · Author response to Decision Letter 1]

7 Nov 2021

Christoph Winkler, Dr.

Academic Editor

PLOS ONE

Manuscript no. PONE-D-21-17584

Manuscript title: Cd9b tetraspanin and Cxcl12a/Cxcr4b have a synergistic effect on the control of collective cell migration

Dear Dr Winkler,

We appreciate that you saw the merit in our research and we were happy to incorporate the minor revisions according to the comments of the reviewers. They are detailed in the point-by-point response below and the modified manuscript has highlighted modified text in red. Previous revisions required additional experiments that were performed by Harsha Mahabaleshwar and Charmaine Ho Min. The main authors who performed the majority of the research are now in positions where they are unable to access the facilities associated with this research. Harsha and Charmaine aided the communication and organisation of the revisions between the authors despite us all being in different continents. They were vital for the latest version of the manuscript and therefore, they were added as additional authors to this manuscript. 

We thank you and the reviewers for your patience and for giving us another chance to meet the publication criteria of PLOS ONE. We hope that you will find the revised manuscript suitable for publication in PLOS ONE.

Sincerely,

Dr. Katherine Marsay

Detailed response to comments:

We would like to thank the reviewers for their constructive comments that helped us to improve the manuscript. As detailed below we modified the manuscript according to these comments. 

Journal Requirements

Comment: Please review your reference list to ensure that it is complete and correct.

Answer: All references were reviewed thoroughly to ensure they were complete, correct and that none had been retracted since the first draft of this manuscript. Two references were added in response to a comment by reviewer 2 as detailed later. Reference 15 was noticed as an incorrect reference and replaced from Charrin et al., 2003 to Ovalle et al., 2007. Reference 42 was noticed to be a duplication of reference 16 and deleted. No other changes to the reference list were required. 

Reviewer #1:

Comment: The title “Augmentation of chemokine signalling by Cd9 tetraspanins facilitates collective cell migration” is missleading as there is no direct data showing that Cd9 enhances Cxcl12/Cxcr4 expresion or activity which in turn facilitates collective cell migration. The authors may consider a more direct title, for instance “Cd9b tetraspanin and Cxcl12a/Cxcr4b convey synergistic effect on the control of collective cell migration”.

Answer: I agree that although our data may provide clues for future research, our current observations on their own are not adequate to suggest a conclusive interaction between Cd9b and the Cxcl12/Cxcr4 pathway in the control of primordium migration. I appreciate the suggested alternative title and would be pleased to use a slightly modified version of this as the new title for our manuscript.

Comment: Line 311: typo “Cxcrb” change to “Cxcr4b”

Answer: Changed. 

Reviewer #2:

Comment: The title should be changed. A subtly different but perhaps less misleading title could be: “Loss of Cd9 exaggerates deficits in collective migration caused by interference with chemokine signaling in the posterior Lateral Line primordium”.

Answer: I agree that although our current hypothesis is that Cd9 modulates collective cell migration of the pLLP through promoting Cxcr4b signaling, we cannot conclusively prove this from our current results. I appreciate the suggested alternative title but have decided to use the following as the new title for our manuscript: “Cd9b tetraspanin and Cxcl12a/Cxcr4b have a synergistic effect on the control of collective cell migration”.

The loss of Cd9b causes a significant delay in collective migration of the posterior lateral line, and also exaggerates deficits in collective migration caused by interference with chemokine signaling. I believe the chosen title describes the phenotype of cd9b KO as being similar to that of loss of Cxcr4b and also highlights the interaction I hypothesized from our preliminary chemokine interactions. I hope you agree but I am happy to change the title again if it is still not clear. 

Comment: Similarly, the abstract should be edited to simply remove or edit the last line: “Loss of both Cd9a and Cd9b sensitized embryos to reduced Cxcr4b and Cxcl12a levels. Together these results provide evidence that Cd9 modulates collective cell migration of the pLLP during zebrafish development.”

OR

“Loss of both Cd9a and Cd9b sensitized embryos to reduced Cxcr4b and Cxcl12a levels. Together these results provide evidence that Cd9 modulates collective cell migration of the pLLP during zebrafish development. One interpretation of these observations is that Cd9 contributes to more effective chemokine signaling.”

Answer: I appreciate the suggestions and agree that the last line of the abstract should be edited to make the conclusions clearer and prevent perpetuating overconfident interpretations of results. The abstract was edited in accordance to the latter suggestion as so: “Together these results provide evidence that Cd9 modulates collective cell migration of the pLLP during zebrafish development. One interpretation of these observations is that Cd9 contributes to more effective chemokine signaling.”

Comment: One additional issue that requires comment. The video of pLLP migration does not show any obvious deficits in migration and migration speed was never assessed or reported. Is it possible that the difference in position of primordium observed by the authors results from a delay in the start of migration. Do the authors have evidence that argues against this? If not, this possibility should also be raised in the discussion.

Answer: The reviewer raises an important point. The delay in primordium migration could either be due to a change in the onset of migration, a change in the speed of migration or a combination of the two. The idea that differences in the observed position of the primordium could be due to a delayed start in migration was added to the discussion. Further experiments that could elaborate on this were also added and involved the addition of two references (44 & 45 - Matsuda et al. 2013, Valdivia et al. 2011). Their publications provide methods that could be used to calculate the speed of primordium migration.

Inserted/edit text is shown below:

Line 387-389: Therefore, the cd9 dKO phenotype appeared to be a migratory defect that could either be due to a change in the onset of migration, a change in the speed of migration or a combination of the two. 

Line 415: We mentioned that the migratory defect could be due to a change in the onset of migration. This could be further investigated by performing claudin b ISHs at the onset and earliest stages of development of the posterior lateral line. On the other hand, to try and determine if Cd9b has a role in the speed of migration, ISHs could also be performed at various time points throughout primordium migration and speed could be calculated in a similar method to previous studies [44,45].

---

## [Editor Report · Decision Letter 2]

9 Nov 2021

Tetraspanin Cd9b and Cxcl12a/Cxcr4b have a synergistic effect on the control of collective cell migration

PONE-D-21-17584R2

Dear Dr. Roehl,

We’re pleased to inform you that your manuscript has been judged scientifically suitable for publication and will be formally accepted for publication once it meets all outstanding technical requirements.

Kind regards,

Christoph Winkler, Dr.

Academic Editor

PLOS ONE
---

## [Editor Report · Acceptance letter]

17 Nov 2021

PONE-D-21-17584R2 

Tetraspanin Cd9b and Cxcl12a/Cxcr4b have a synergistic effect on the control of collective cell migration 

Dear Dr. Roehl:

I'm pleased to inform you that your manuscript has been deemed suitable for publication in PLOS ONE. Congratulations! Your manuscript is now with our production department. 

Kind regards, 

on behalf of

Dr. Christoph Winkler 

Academic Editor

PLOS ONE